# Few-Shot Image Classification: Current Status and Research Trends

**Ying Liu** [1,*] **, Hengchang Zhang** [1]**, Weidong Zhang** [1]**, Guojun Lu** [2]**, Qi Tian** [3] **and Nam Ling** [4]

1   Center for Image and Information Processing, Xi'an University of Posts and Telecommunications, Xi'an 710121, China; z269998166@163.com (H.Z.); chluzhre@126.com (W.Z.)
2   School of Engineering, Information Technology and Physical Sciences, Federation University Australia, Gippsland 3841, Australia; guojun.lu@federation.edu.au
3   Huawei Technologies Co., Ltd., Shenzhen 518000, China; tian.qi1@huawei.com
4   Department of Computer Science and Engineering, Santa Clara University, Santa Clara, CA 950053, USA; nling@scu.edu
*   Correspondence: liuying2018@xupt.edu.cn or liuying_ciip@163.com

**Abstract:** Conventional image classification methods usually require a large number of training samples for the training model. However, in practical scenarios, the amount of available sample data is often insufficient, which easily leads to overfitting in network construction. Few-shot learning provides an effective solution to this problem and has been a hot research topic. This paper provides an intensive survey on the state-of-the-art techniques in image classification based on few-shot learning. According to the different deep learning mechanisms, the existing algorithms are divided into four categories: transfer learning based, meta-learning based, data augmentation based, and multimodal based methods. Transfer learning based methods transfer useful prior knowledge from the source domain to the target domain. Meta-learning based methods employ past prior knowledge to guide the learning of new tasks. Data augmentation based methods expand the amount of sample data with auxiliary information. Multimodal based methods use the information of the auxiliary modal to facilitate the implementation of image classification tasks. This paper also summarizes the few-shot image datasets available in the literature, and experimental results tested by some representative algorithms are provided to compare their performance and analyze their pros and cons. In addition, the application of existing research outcomes on few-shot image classification in different practical fields are discussed. Finally, a few future research directions are identified.

**Keywords:** few-shot learning; transfer learning; meta-learning; data augmentation; multimodal

## 1. Introduction

Image classification aims to distinguish different types of images according to semantic information, which is an important application in computer vision. Prior to the emergence of deep-learning techniques, local features, such as Bag of Words (BoW) [1], Scale-Invariant Feature Transform (SIFT) [2], Histogram of Oriented Gradient (HOG) [3], and Local Binary Pattern (LBP) [4], were commonly used tools in image classification tasks. However, these traditional methods heavily rely on manual design, which is not only computationally complex but also inefficient. On the other hand, traditional image classification algorithms extract single features of an image as input, such as texture features [5], which can only represent partial information of an image and cannot accurately describe the image, making it difficult to produce good results for image classification tasks. These classical methods are generally for a specific identification task, and the size of the data is not large and the generalization ability is poor, making it difficult to achieve an accurate recognition effect in practical applications [6].

In recent years, with the continuous development of artificial intelligence and the proposal of large-scale labeled datasets [7] and deep neural network structures [8], many researchers have trained models with excellent recognition performance by convolutional

neural networks (CNN) using large-scale labeled data [9–11]. Model training relies on a huge amount of data. However, it is difficult to collect massive data samples in practical application fields, such as medicine and military. Besides, sample labeling is time-consuming and labor-intensive, and the training equipment also requires huge investment. A model without sufficient training data is prone to overfitting. Few-shot learning (FSL) [12,13] can effectively solve this problem. The concept of few-shot learning was first proposed by Feifei Li et al. in 2003, pointing out that the key problem of few-shot learning is how to use the learned knowledge to learn a new category [14]. This technique solves the long-standing problem with the need for large and extensive datasets. For training samples, few-shot learning generally only needs to learn features of a small number of labeled images to classify new test images. At present, few-shot learning has been widely used in many image processing tasks, such as image recognition [15], image segmentation [16], image classification, and retrieval [17–19]. In addition, the research of few-shot image classification also has high application value. In fields such as medical [20] and public security [21], it is difficult to collect large-scale labeled data, making deep learning models perform poorly. Few-shot learning can effectively alleviate the problem wherein some high-performance models cannot generalize in new classes due to the small amount of training data; this enables these high-performance models to be applied to more fields. At present, some scholars have reviewed few-shot learning [22,23]. Zhao et al. [22] specifically introduced the research progress of few-shot learning models and algorithms in accordance with the methods based on model fine-tuning, data augmentation, and transfer learning. Wang et al. [23] conducted an extensive literature review on few-shot learning and organized it into a unified taxonomy from the perspective of data, model, and algorithm. However, there is very little literature in academia reviewing few-shot image classification [24].

In the early stage of the research, few-shot learning is based on the Bayesian framework, and the class probability reasoning of the sample is obtained by combining the model parameters with the prior probability and the posterior probability [12]. With the development of deep learning and the evolution of neural network architecture, researchers have introduced neural network models to solve the problem of few-shot image classification. In recent years, most of the existing few-shot learning methods have adopted the deep learning technique. The difference between this paper and the aforementioned review articles [22,23] is that this paper focuses on the existing methods for few-shot image classification, whereas literature [22,23] discuss few-shot learning in different tasks such as image segmentation, target detection, natural language processing, etc. The main contributions of this paper are as follows: (1) based on an intensive survey on recent literature, this paper divides the existing few-shot image classification algorithms into four categories: transfer learning-based, meta-learning-based, data augmentation-based, and multimodal-based. (2) The few-shot image datasets available in the literature are summarized. (3) Performance of representative algorithms from each category is compared based on experimental results. (4) A list of future research directions in few-shot image classification is identified.

The structure of this paper is as follows: Section 2 defines the few-shot image classification tasks and summarizes the commonly used datasets in this field; in Section 3, the existing few-shot image classification algorithms are classified into four categories: transfer learning-based, meta-learning-based, data augmentation-based, and multimodal-based; Section 4 compares the experimental results of some representative algorithms and discusses the pros and cons of various methods; Section 5 describes the application of few-shot image classification in different practical fields; Section 6 discusses the research trends of few-shot image classification; Section 7 concludes this paper.

## 2. Definition and Datasets

### 2.1. Few-Shot Image Classification Definition

A few-shot image classification (FSIC) task is generally termed as a *N-way K-shot* [25] problem. A training set of few-shot learning contains many categories, and there are multiple samples in each category. In the training phase, *N* categories of image samples

are randomly selected from the training set, and $K$ samples (a total of $N \times K$ images) are selected from each category as the support set. Then, a small number of samples from the remaining data of each of these $N$ categories are selected as the prediction object of the model, also known as the query set. If $K$ is very small (usually $K < 10$), the classification task is called the few-shot image classification; when $K = 1$, it becomes a one-shot image classification task; when $K = 0$, it is a zero-shot image classification task. Few-shot learning generally adopts the episode training mechanism. An episode contains a support set and a query set. After learning from the support set, the performance of the model is verified on the query set. Therefore, an episode corresponds to a few-shot learning task. The goal of a few-shot image classification task is to accurately classify the images in the query set based on the existing support set, that is, the model is required to learn how to distinguish these $N$ categories from $N \times K$ samples.

### 2.2. Datasets

This paper summarizes the few-shot learning datasets commonly used in academia and divides them into simple image datasets, complex image datasets, and special image datasets according to different sample data types. Simple image datasets include the Omniglot dataset [26] and the MNIST dataset [27], both of which have simple image content, such as handwritten characters and handwritten numbers, which are easy to classify. Complex image datasets include the miniImageNet dataset [25], tieredImageNet dataset [28], CIFAR-100 dataset [29], and Caltech101 dataset [30], which have richer and more complex image categories, such as images of people, animals, cars, etc., increasing the difficulty of the classification task. The image contents of special image datasets are relatively similar with large intra-class differences and small inter-class differences, and the image samples of the dataset are all a particular kind of object. The special image dataset includes the CUB-200 dataset [31] and the CIIP-TPID dataset [32]. Table 1 summarizes the existing commonly used few-shot learning datasets.

The MNIST dataset consists of 60,000 training samples and 10,000 test samples, each of which is a 28 × 28 pixel image of a handwritten number.

The Omniglot dataset is composed of 1623 handwritten characters from 50 different languages. Each character has 20 different handwritings, which is equivalent to 1623 categories with 20 samples per category.

The miniImageNet dataset contains 60,000 color images in 100 categories, with 600 samples in each category, and the size of each image is 84 × 84. Among them, the training set, validation set, and test set contain 64 categories, 16 categories, and 20 categories, respectively.

The tieredImageNet dataset contains 608 categories with a total of 779,165 images. Among them, the training set, validation set, and test set contain 351 categories, 97 categories and 160 categories, respectively.

The CIFAR100 dataset has 100 classes, each class has 600 color images of size 32 × 32, 500 of which are used as a training set and 100 as a test set.

The Caltech101 dataset is composed of 101 categories of object pictures, including animals, industrial products, pizza, and human faces. There are 40 to 800 pictures in each category, and the size of each picture is 300 × 200 pixels. The total number of pictures in this dataset is 9146.

The full name of the CUB-200 dataset is Caltech-UCSD Birds-200 dataset, which contains 6033 images of 200 species of birds.

The CIIP-TPID dataset was established by the Center for Image and Information Processing (CIIP) of Xi'an University of Posts and Telecommunications (XUPT). According to the actual needs of public security and traffic police, a total of 11,040 images of 69 types of tire pattern images under different environments, illuminations, and angles have been established in this dataset. Each category includes 80 tire surface patterns and 80 tire indentation images. The tire pattern mixing database is the largest database published in this field so far and has been used as the official dataset of the Multimedia Grand Challenge-Fine Grained Vehicle Footprint Recognition competition of the ACM Multimedia Asia

2019 conference [33] and the Grand Challenge-Few-Shot Learning for vehicle footprint recognition competition of the ICME 2021 conference [34].

**Table 1.** The existing commonly used few-shot learning datasets.

| Types | Datasets | Source | Categories | Images | Examples |
|-------|----------|--------|-----------|--------|----------|
| Simple image datasets | Omniglot | New York University | 1623 | 32,460 | |
| | MNIST | New York University | 10 | 70,000 | |
| Complex image datasets | miniImageNet | Google DeepMind team | 100 | 60,000 | |
| | tieredImageNet | University of Toronto | 608 | 779,165 | |
| | CIFAR-100 | University of Toronto | 100 | 60,000 | |
| | Caltech101 | California Institute of Technology | 101 | 9146 | |
| Special image datasets | CUB-200 | California Institute of Technology | 200 | 6033 | |
| | CIIP-TPID | Xi'an University of Posts and Telecommunications | 69 | 11,040 | |

## 3. Few-Shot Image Classification Algorithms

According to different learning paradigms, this paper summarizes the existing few-shot image classification algorithms into four categories: transfer learning-based, meta-learning-based, data augmentation-based and multimodal-based. (1) The transfer learning-based method aims to apply the knowledge learned in a certain field or task to different but related fields or problems. According to different mechanisms, transfer learning-based methods can be further divided into instance-based, feature-based, and fine-tuning-based methods. (2) The meta-learning-based method employs previous knowledge and experience to guide the learning of new tasks so that the model has the ability to learn to learn. According to different mechanisms, meta-learning-based methods are subdivided into model-based, optimization-based, and metric-based methods. (3) the data augmentation-based method expands the training set through various processing of existing data to solve the problem of insufficient training data. According to different augmentation ways, it can be divided into data generation-based and feature enhancement-based methods. (4) The

multimodal-based method learns better feature representations by eliminating redundancy between modalities by exploiting the complementarity between multiple modalities. The multimodal-based method is implemented in two ways: knowledge transfer-based and metric-based methods. The existing few-shot image classification algorithms are summarized in Figure 1.

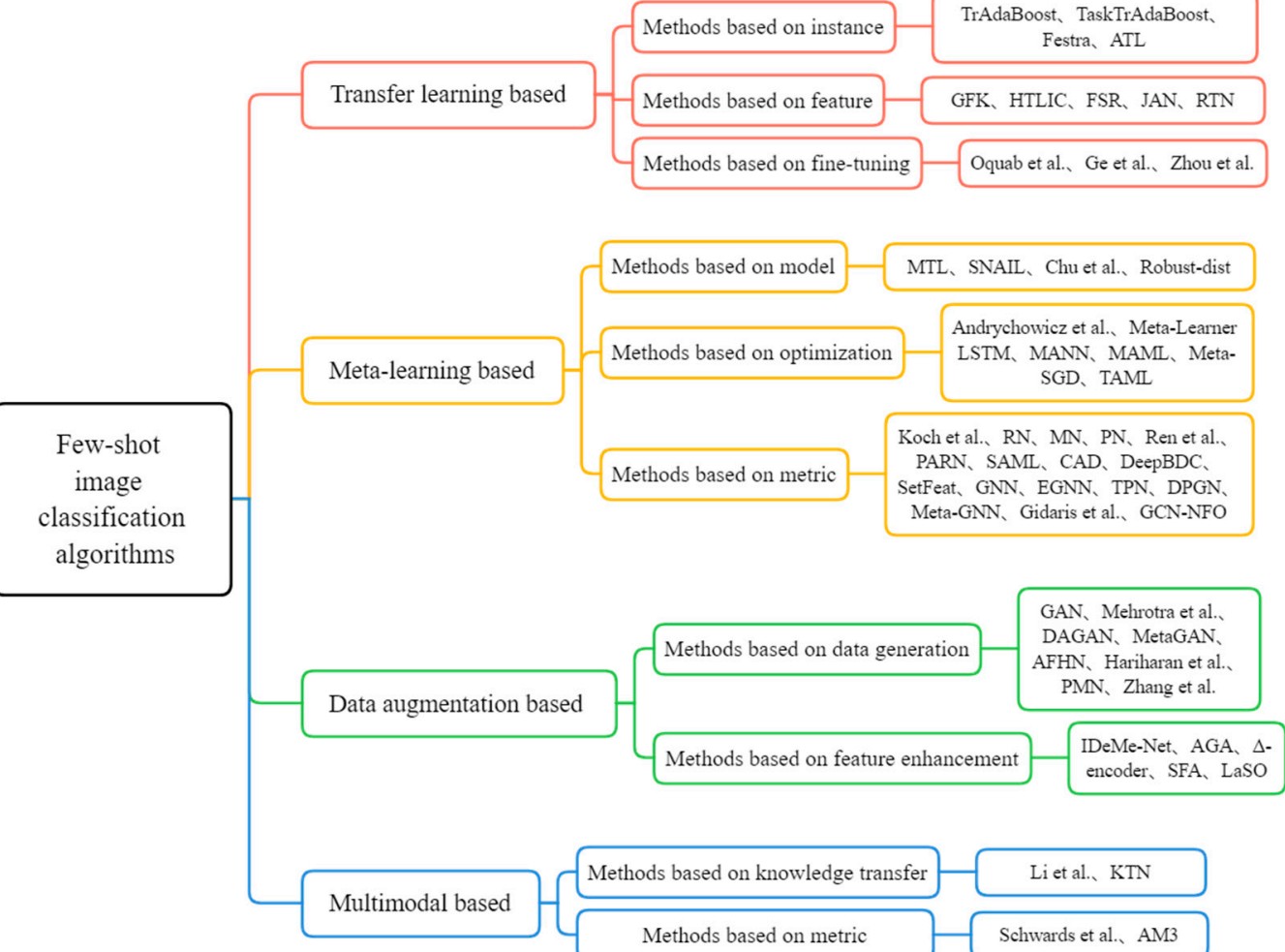

**Figure 1.** An overview of image classification algorithms based on few-shot learning.

### 3.1. Transfer Learning-Based Methods

The main difficulty of few-shot learning lies in how to optimize the model when new data classes appear but each class has no labeled training samples. Considering that there are sufficient-known tagged data from related fields, this problem can be solved by transfer learning (TL). The schematic diagram of transfer learning is shown in Figure 2. As a machine learning method, transfer learning transfers the useful prior knowledge in the source domain to the target domain, which is conducive to few-shot learning. Using the prior knowledge from the source domain, the performance of the learning tasks in the target domain can be improved even in the case of fewer samples. In some cases, the violent transfer may fail when the source domain and the target domain are not related to each other. In the worst case, it may even damage the learning performance of the target domain, which is called negative transfer [35]. Generally, this scenario is suitable for transfer learning when the data in the source domain is sufficient and the data in the target domain is small. According to the different mechanisms and technical means in the process of transfer learning, this paper divides the transfer learning methods into instance-based, feature-based, and fine-tuning-based methods.

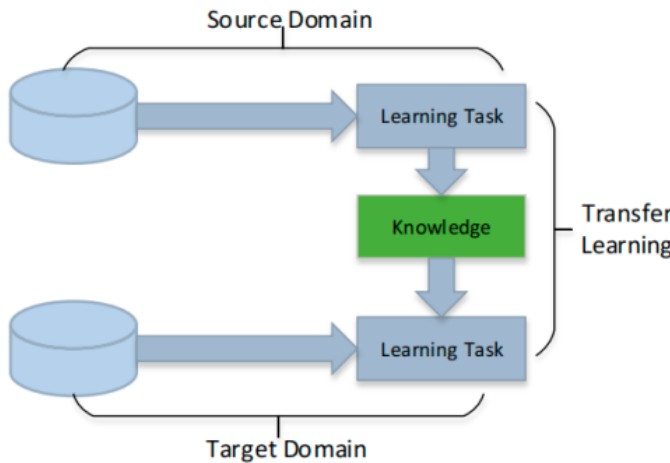

**Figure 2.** The learning process of transfer learning.

3.1.1. Instance-Based Methods

In transfer learning, there is a high risk of generalization when a certain class of samples has a high probability of occurring in the target domain and a low probability of occurring in the source domain. Instance-based transfer learning is to find a way to weight the input sample features. Instance-based transfer learning studies how to select examples from the source domain that are useful for training in the target domain, such as effectively assigning weights to labeled data instances in the source domain, this ensures that the instance distribution in the source domain is close to that in the target domain, so as to establish a reliable learning model with higher classification accuracy in the target domain. The schematic diagram of instance-based transfer learning is shown in Figure 3.

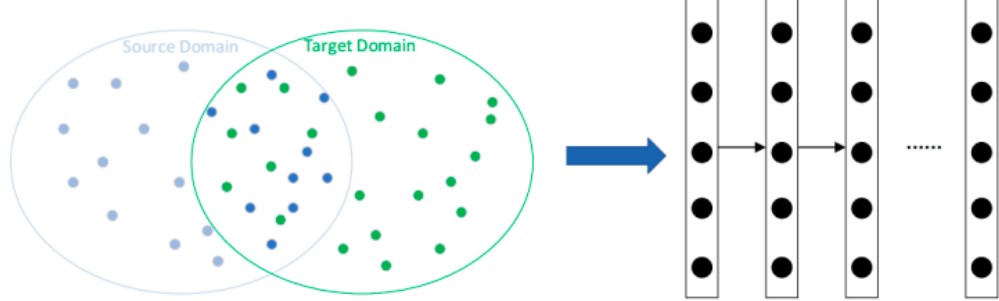

**Figure 3.** Schematic diagram of instance-based transfer learning.

The TrAdaBoost method proposed by Dai et al. [36] uses an AdaBoost-based approach to filter out instances in the source domain that are different from the target domain. The instances are reweighted in the source domain to form a distribution similar to the target domain. Finally, the model is trained by using reweighted instances from the source domain and original instances from the target domain. Since it is difficult to measure the correlation between any independent source domain and target domain, Yao et al. [37] extended TrAdaBoost and proposed a boosting method called TaskTrAdaBoost to minimize the impact of negative transfer of irrelevant source domains for knowledge transfer. This method can promote rapid retraining of new target domains.

In [38], a transfer learning method called Festra was proposed to deal with the problem of interregional sandstone microscopic image classification. This method includes both feature selection and E-TrAdaBoost; the latter combines the technique of feature and instance transfer. The purpose of feature selection is to filter out the features with large differences between the target domain and the source domain, while E-TrAdaBoost aims to reduce the difference between slice images collected in different regions. Therefore,

labeled instances from multiple regions can be used to train high-quality classifiers to make predictions in the target domain.

In order to solve the impact of negative transfer, Liu et al. [39] proposed the Analogical Transfer Learning (ATL), which follows the analogy strategy. The algorithm first learns a modified source hypothesis, and then transforms the modified source hypothesis and the target hypothesis (trained with only a small number of samples) into an analogous hypothesis. The experimental results show that the algorithm effectively controls the occurrence of negative transfer on the two levels of instance and hypothesis and has a better generalization capability.

### 3.1.2. Feature-Based Methods

In transfer learning, it is usually difficult to have a good overlap between the feature space of source domain and target domain, so it is necessary to find useful features on the basis of the feature space. The feature-based transfer learning algorithms focus on how to find the common feature representations between the source domain and the target domain, and then use these features for knowledge transfer, which can be a good solution to this problem. Gong et al. [40] proposed a domain adaptation technique Geodesic Flow Kernel (GFK), which reduces the difference of edge distribution by finding a low-dimensional feature space. In addition, when there are multiple source domains, a domain level metric ROD is proposed, which calculates the distance between each source domain and the target domain for selecting the appropriate source domain transfer.

In transfer learning, a large amount of unlabeled heterogeneous source data can be used in some cases to improve the prediction performance of a specific target learner. Zhu et al. [41] proposed a Heterogeneous Transfer Learning Image Classification (HTLIC) method, which uses a large amount of available unlabeled source domain data to create a common potential feature input space in order to improve the prediction performance of the target classifier. Experiments show that this method can be effectively used for image classification tasks. Feuz et al. [42] proposed a new heterogeneous transfer learning method called Feature-Space Remapping (FSR), which transfers knowledge between domains with different feature spaces and associates features in different feature spaces by constructing meta features.

Deep networks have been successfully applied to learn transferable features to adapt the model from the source domain to different target domains. The Joint Adaptation Networks (JAN) proposed by Long et al. [43] is able to learn the transmission network by aligning the joint distribution of multiple domain-specific layers across domains based on the joint maximum average difference criterion. In addition, the author also proposed a new deep network unsupervised domain adaptive method called Residual Transfer Network (RTN) [44], which can realize the end-to-end learning of adaptive classifiers and transferable features.

### 3.1.3. Fine-Tuning-Based Methods

In the research of deep learning, there are usually few large-scale datasets that can be used to train neural networks, resulting in poor classification results. Fine-tuning-based methods can avoid the need for large-scale training datasets, so as to improve the classification effect. The transfer learning method based on fine-tuning first trains a model on another large dataset, and then adopts the weights obtained from the training as the initial weights of the new task (small dataset), and finally retrains the model with a smaller learning rate. The advantage of this method is that the number of training parameters can be reduced, which is beneficial to overcome overfitting.

CNN has achieved excellent results in the field of computer vision, but the training of CNN models often requires a lot of labeled data and its performance on small datasets will be worse. To solve this problem, Oquab et al. [45] adopted a fine-tuning method. First, the traditional CNN model was pre-trained on large datasets such as ImageNet, and then fine-tuned for specific tasks. This method modified the entire pre-training framework

by removing the softmax activation layer and adding two adaptive layers with learnable parameters while the rest of the parameters are frozen.

Ge et al. [46] proposed a selective joint fine-tuning transfer learning method, which introduces some tasks with sufficient training sets into the current task for joint training. The introduced auxiliary task selects the part of the images whose underlying features are similar to the training set of the current main task. Using additional data that can be directly obtained to assist the training of the main task; this method can effectively reduce the risk of overfitting caused by insufficient training data and improve the classification accuracy of the model.

At present, many few-shot learning methods based on transfer learning are pre-trained on the basic dataset, and then fine-tuned on the new few-shot dataset. However, how to choose the best basic dataset for pre-training is still a difficult problem. Zhou et al. [47] proposed an algorithm to improve few-shot learning by guiding the selection of basic categories. Specifically, first, the concept of similarity ratio (SR) was introduced to describe the relationship between the category selection of basic datasets and the classification effect on the new dataset; then, the problem of base class selection is further expressed as a submodel optimization problem based on SR; finally, the optimal solution of this problem is obtained by a greedy algorithm.

### 3.2. Meta-Learning-Based Methods

Deep learning has achieved great success and has become a practical method in many applications, such as computer vision and natural language processing. However, this relies heavily on a large amount of labeled training data. As a standard method to solve the problem of few-shot learning, meta-learning tries to learn how to learn. The goal of meta-learning is to enable models, especially deep neural networks, to learn from only a small number of data samples how to take on new tasks. Meta-learning is essential for machine intelligence and also very challenging. According to the different mechanisms, this paper divides meta-learning into model-based, optimization-based, and metric-based methods. Figure 4 shows an example of the application of meta-learning in the field of image classification.

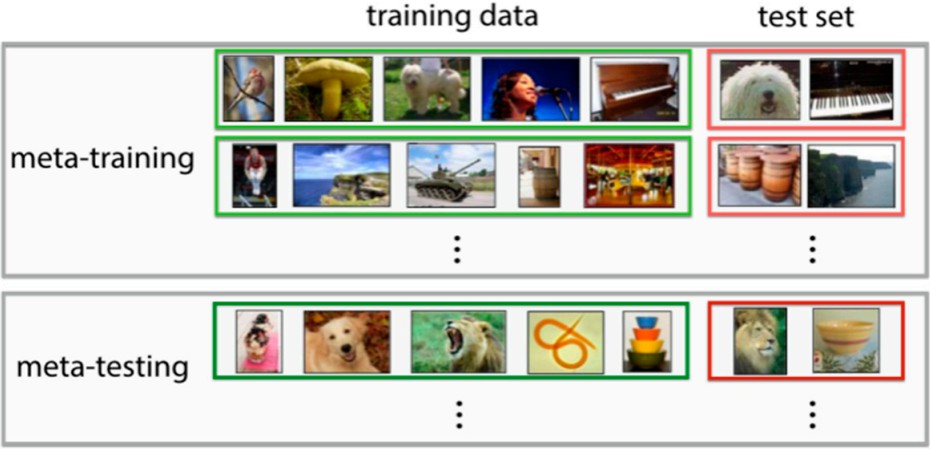

**Figure 4.** An example of applying meta-learning in the field of image classification.

### 3.2.1. Model-Based Methods

The model-based meta-learning method aims to generate some parameters of the model by using the general knowledge learned in different tasks so that the model can adaptively solve the corresponding tasks, thereby improving the performance of few-shot learning classification tasks.

Sun et al. [48] proposed a new meta-learning method called Meta-Transfer Learning (MTL) by combining the advantages of transfer learning and meta-learning. In MTL, the scaling and translation parameters are introduced to adjust the weight parameters to meet

the needs of new tasks. This method avoids updating the weight parameters of the entire network and reduces the problem of overfitting. In addition, deep convolution neural network can be used to improve the ability of feature representation. Finally, the course learning method is adopted to train the network from simple to difficult, which effectively improves the loss convergence speed and the classification effect.

SNAIL [49] is a general meta-learning architecture, which is composed of interleaved time convolution and causal attention layer. The convolution network learns the general feature vector of the training samples to aggregate the information from past experience. The causal attention layer selects the information from the collected experience to be popularized to the new task, which can effectively complete the few-shot learning tasks.

Chu et al. [50] proposed a reinforcement learning model based on the maximum entropy block sampling algorithm to solve the few-shot classification tasks. The model divides a picture into several blocks and then forms a block sequence for learning. This method can guide the network to select the valuable parts of the picture for observation, without wasting energy in the background area, which improves the learning efficiency of the model to a certain extent.

Dvornik et al. [51] proposed a few-shot learning algorithm based on model ensemble, called Robust-dist, which integrates multiple models together and calculates the final result by voting or averaging the output results of each model. The algorithm uses ensemble learning to reduce the divergence between classifiers and improve the effect of few-shot learning.

### 3.2.2. Optimization-Based Methods

In few-shot image classification tasks, the learner is usually overfitting due to the small number of training samples, and in the training process, the learner is usually trained for millions or even tens of millions of iterations before converging in order to achieve a better result. These problems not only affect the performance of the learner, but also the classification efficiency of the model. The optimization-based meta-learning method is an important branch in the field of few-shot learning. This type of algorithm tries to obtain a better initialization model or gradient descent direction through meta-learning and optimizes the initialization parameters of the learner by means of a meta-learner so that the learner can converge faster in the corresponding task and can learn fast with only a small number of samples. Some existing methods use an additional neural network, such as Long Short-Term Memory (LSTM), as a meta-learner to train the model. In [52], a meta-learner is developed, based on LSTM, and shown how to transform the design of an optimization algorithm into a learning problem. Ravi et al. [53] proposed another LSTM-based meta-learner to learn appropriate parameter updates and general initialization of the learning model. Compared with LSTM, Santoro et al. [54] proposed the Memory-Augmented Neural Network (MANN), which trains the Neural Turing Machine (NTM) [55] as a meta-learner. This is a neural network with enhanced memory capabilities. The displayed external memory module is used to retain the sample feature information, and the meta-learning algorithm is used to optimize the reading and writing process of NTM. The writing process closely associates the feature information with the corresponding label, and the reading process accurately classifies the feature vector, finally realizing effective few-shot classification and regression tasks.

Finn [56] proposed a new meta-learning algorithm called Model-Agnostic Meta-Learning (MAML). First, the network is trained to have the ability of common feature extraction, and then on this basis, the network is further trained to quickly adapt to new tasks, that is, a parameter initialization state with high sensitivity is obtained through learning, in which a small change of parameters can greatly improve the loss function. This method is considered model agnostic because it can be directly applied to any learning model trained by the gradient descent process.

The advantage of MAML is that it uses meta-learning to obtain a better initialization parameter. On this basis, better results can be obtained by fine-tuning on a small number

of samples. However, the disadvantage is that the capacity of the model is limited as only the initialization parameters are learned, which is generally only suitable for shallow networks. The LSTM-based meta-learner method uses the LSTM network as the outer network to learn the optimization parameters of the inner network. This method has a large model capacity, but it is not practical because the LSTM training process is complex and the convergence speed is slow. Based on this, Li et al. [57] proposed a compromise method called Meta-SGD, which follows the method MAML that only needs the same network structure for internal level training and external level training, respectively. Through meta-learning, the initialization parameters; learning rate; and update direction are learned at the same time, then the trained model can be easily fine-tuned to adapt to new tasks. Compared with MAML, the model capacity of this algorithm has been improved. Compared with LSTM, the training difficulty of this algorithm has been significantly reduced.

A potential problem of the optimization-based meta-learning method is that the model tends to have a preference for training tasks during the training process, which may lead to the decline of its generalization ability. To solve this problem, Jamal [58] proposed a method called Task-Agnostic Meta-Learning (TAML), which is further improved on MAML. On the original basis, it explicitly requires that the parameters of the model have no preference for different tasks through regularization, so as to improve the generalization ability of the model to new samples.

### 3.2.3. Metric-Based Methods

Metric learning is a method of spatial mapping, which can learn a feature space where all data are transformed into a feature vector and the distance between the feature vectors of similar samples is smaller than that of dissimilar samples for data distinction.

By comparing the similarity between data samples, several specialized models for metric-based meta-learning have emerged [59–61], especially for few-shot classification tasks. Specifically, Koch et al. [59] first introduced the Siamese Neural Networks for few-shot classification tasks in 2015. In addition, Sung et al. [60] proposed a conceptually simple, flexible, and general few-shot learning framework called the Relation Network (RN), which is trained end-to-end from scratch. The classifier in RN learns several examples from each category to train the network in an end-to-end manner and adjusts the embedding and distance metrics to achieve effective few-shot image classification. Vinyals et al. [25] proposed the Matching Networks (MN), which learns an embedding function and uses the cosine distance in the attention kernel to measure similarity. Figure 5 shows the architecture of MN. Snell et al. [61] proposed the Prototypical Networks (PN), which maps examples to a p-dimensional vector space so that the examples of a given output category are close to each other. Then, it calculates the prototype (average vector) for each category. The new sample will be mapped to the same vector space, and the distance metric will be used to create a Softmax among all possible categories to classify the sample. On this basis, Ren et al. [28] proposed three improved models to extend the PN to semi-supervised learning.

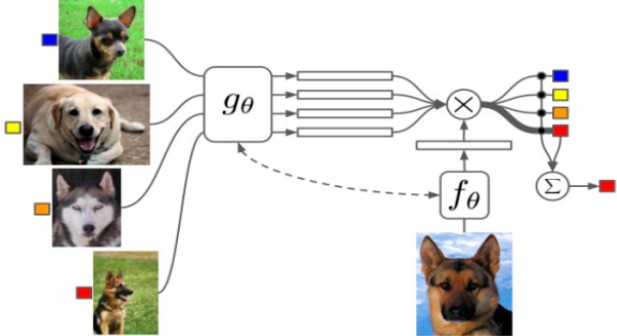

**Figure 5.** The structure of MN, where $g_\theta$ and $f_\theta$ are the coding functions of training data and test data, respectively.

The ordinary CNN feature extraction networks will only get a high response at the location of the target object. In this case, if the image target object in the support set and the query set are not in the same location, the obtained feature map will not correspond well. In order to solve the above problems, Wu et al. [62] made improvements based on RN and proposed Position-Aware Relation Networks (PARN). Specifically, first, Deformable Feature Extractor (DFE) is used to extract more useful features in feature extraction, so as to improve the effect of similarity comparison. Then, the Dual Correlation Attention (DCA) mechanism is proposed to aggregate the correlation information between any two pixels between the query set image and the support set image. This method uses fewer parameters to achieve better results.

In the existing meta-learning methods, based on metric, directly calculating the distance between the query set image and support set image may cause ambiguity, as the main object can be located anywhere on the image. In order to solve this problem, Hao et al. [63] proposed a Semantic Alignment Metric Learning (SAML), which aligns semantic-related main objects through the "collect and select" strategy. Firstly, a relation matrix is calculated to "collect" the distance between each local region pair of a 3D tensor extracted from the query set image and the average tensor of the support set image, and then the attention mechanism is used to "select" and pay attention to the semantically related local region pairs, and finally the weighted relationship matrix is mapped to its corresponding similarity score by multi-layer perceptron.

Chikontwe et al. [64] proposed a strategy to cross-attend and re-weight discriminative features (CAD) for few-shot image classification. A single shared module is introduced to produce a pooled attention representation of features by calculating the mutual attention scores. This method can effectively re-weight features to boost performance and generalize better for cross-domain tasks.

Xie et al. [65] proposed Deep Brownian Distance Covariance (DeepBDC), where the BDC matrix is calculated to represent the input image and a more accurate similarity of a pair of images can be obtained by the inner product of the corresponding BDC matrixes. This greatly improves the performance of few-shot image classification.

Afrasiyabi et al. [66] introduced SetFeat for set feature extraction. A set of M feature vectors is extracted from the images, and another set of M feature vectors is generated by embedding a shallow self-attention based mapper at different stages of the network. The set-to-set matching metric is used to establish the similarity between images in the set-feature space during the training and inference to classify few-shot images.

The graph is composed of edges and nodes. The construction process of the graph includes the generation of edges and the updating of nodes. The representation of edges is the measurement of the relationship between nodes in the graph. In recent years, the Graph Neural Network (GNN) based on meta-learning has been paid more and more attention in few-shot learning. GNN can be attributed to embedding learning. By mapping the samples into the feature space and then measuring the relationship between samples, GNN classifies the samples according to the measurement distance. Therefore, the graph neural network is essentially a metric-based method to achieve few-shot learning. The nodes of the graph neural network can represent a single sample in the training set, and the edges can represent the correlation between the samples, relying on the information transfer between the nodes in the graph to capture the dependency relationship in the graph, and have strong representation ability.

At present, some existing few-shot learning methods with GNN represent and learn the intra-class sample relationships or inter-class sample relationships through the nodes and edges in the graph and iteratively update the graph. Garcia et al. [67] proposed an end-to-end GNN, which uses GNN to directly predict the category of unknown samples. Firstly, feature extraction is carried out for the support set and query set, and then these sample features and corresponding labels are spliced as the input of the graph network. In the process of iteratively updating the graph, the relationship between intra-class samples

and inter-class samples is implicitly constructed, and the classification task is completed after the update. The proposed GNN structure is shown in Figure 6.

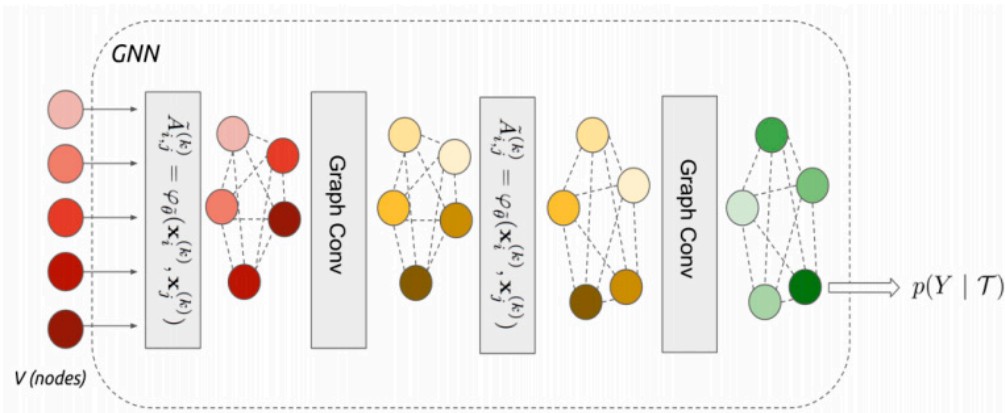

**Figure 6.** Overview of the proposed GNN.

In contrast, the Edge-Labeling Graph Neural Network (EGNN) proposed by Kim et al. [68] learns to predict the edge labels on the graph instead of the node labels and explicitly models the similarity and dissimilarity between samples based on prior knowledge. Liu et al. [69] proposed a Transductive Propagation Network (TPN), in which node features are obtained through deep neural networks. The network uses the sample features extracted by the convolution block as the input of the graph network and transfers the label from the support set samples to the query set samples according to the output feature of the graph network. The network architecture diagram of TPN is shown in Figure 7. The Distribution Propagation Graph Network (DPGN) proposed by Yang et al. [70] does not only focus on the relationship between samples but introduces a graph network method for label propagation through sample distribution to integrate the relationship between instance level and distribution level, so as to transmit information between point graph and distribution graph to realize image classification.

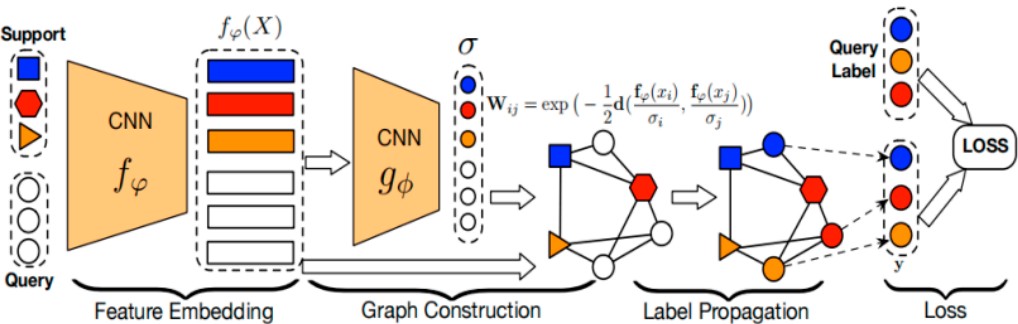

**Figure 7.** The network architecture diagram of TPN.

The Meta-GNN [71] trains multiple similar few-shot learning tasks to obtain the prior knowledge of the classifier, and then uses a new class with a small number of label samples to classify nodes. Gidaris et al. [72] proposed a few-shot learning algorithm using graph neural networks. The algorithm can be updated based on the weight vector of the basic category and a small number of new samples to obtain the category weight vector corresponding to the new sample, which cannot only identify the new sample, but also retain the classification ability of the basic category. In order to quickly update the weight parameters, the Denoising AutoEncoders (DAE) are introduced into the GNN. Gaussian noise is added to the initial weight vector, and then the vector is restored and reconstructed. The direction of weight update is guided by the difference between the reconstructed vector and the initial vector.

The representation ability of graph node features is an important factor affecting the learning performance of the graph convolution network. Liu et al. [73] proposed an improved graph convolution network model called GCN-NFO, which uses the cross attention mechanism to associate the image features of the support set and the query set and extracts more representative and discriminative salient region features than the initialization features of graph nodes through information aggregation. The optimized graph node features transmit information through the graph network, which implicitly strengthens the similarity of intra-class samples and the difference of inter-class samples, thereby enhancing the learning ability of the model.

### 3.3. Data Augmentation-Based Methods

The fundamental problem of few-shot learning is that the number of samples is too small, which leads to lower sample diversity and makes the model prone to overfitting. When the amount of data is limited, the number and category of samples in the dataset can be expanded by the methods based on data augmentation to improve sample diversity and prevent overfitting of few-shot learning models during training. At the beginning of the development of deep learning, data augmentation usually generates new samples by performing some transformations on the sample data. These transformations include operations such as rotation, deformation, scaling, cropping, and color transformation. With the continuous development of few-shot learning, more advanced data augmentation methods are constantly being proposed. In this paper, according to the different ways of enhancement, we will divide them into those methods based on data generation and those based on feature enhancement, and then we will introduce them in detail.

### 3.3.1. Data Generation-Based Methods

The methods based on data generation aim to generate new sample data for few-shot categories for the purpose of data augmentation. Although data augmentation can expand the sample data in the dataset and reduce the overfitting problem in few-shot learning to a certain extent, the transformation mode is limited due to the small amount of sample data. Although the training effect has been improved to a certain extent, the overfitting problem cannot be completely solved.

Goodfellow et al. [74] proposed the famous Generative Adversarial Nets (GAN), which consists of a generator and a discriminator. The task of this model is to train two competing networks for dynamic games. The generator generates an image that is as similar as possible to the real image and aims to prevent the discriminator from judging whether the image is a real image or an image generated by the generator. The discriminator distinguishes the image generated by the generator from the real image as accurately as possible. Mehrotra et al. [75] further proposed to generate samples for specific tasks to expand the training set and combined GAN with a few-shot classification network to make the generated samples more suitable for few-shot learning.

Based on the image generation adversarial model, Antoniou et al. [76] proposed the Data Augmentation Generative Adversarial Networks (DAGAN). The model obtains image data from the source domain and inputs it into the encoder to be projected into a lower-dimensional vector. The converted random vector is connected to the decoder to generate an enhanced image. The MetaGAN model proposed by Zhang et al. [77] combines GAN with part of the classification network for training. This method can help the few-shot classifier learn a clearer decision boundary, thereby helping to improve the performance of few-shot learning.

Li et al. [78] proposed the Adversarial Feature Hallucination Networks (AFHN), which uses the conditional Wasserstein Generative Adversarial Networks (cWGAN) to generate samples for dataset expansion. By adding a classification regularizer and an anti-collapse regularizer, the discrimination ability and diversity of generated samples are improved, so that they can be applied to few-shot learning. The framework of AFHN is shown in Figure 8.

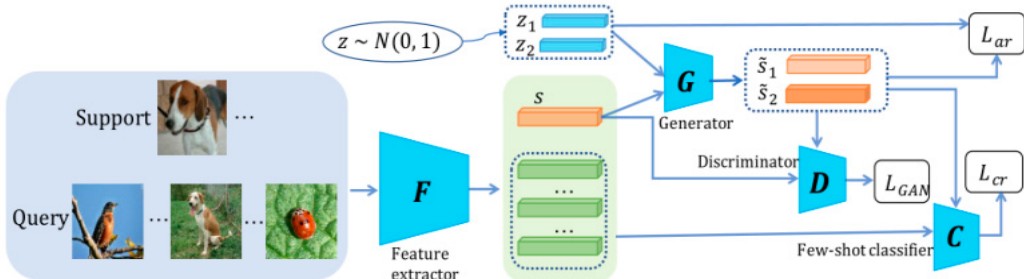

**Figure 8.** The framework of AFHN.

In addition to the above GAN-based methods, there are some other methods that can also solve the problem of insufficient data. Few-shot learning can identify new object categories from a few samples. In order to ensure sample diversity, Hariharan et al. [79] proposed a data augmentation method based on few-shot learning of complex images. This method transforms the two sample feature vectors of the same category, and then applies them to the sample feature vector of the new category to generate a new sample and adds it to the training set of the new category. Wang et al. [80] proposed the Prototype Matching Networks (PMN), in which the image synthesizers, feature extraction networks, and classifiers are combined into one network for end-to-end training. The loss of classification is used to guide the training of the image synthesizer so that it can output images that can meet the needs of classification. Based on the relational network, Zhang et al. [81] used a salient target detection algorithm to segment the image into foreground and background, and then merged the foreground and background of different pictures to form more composite images, so as to expand the dataset.

### 3.3.2. Feature Enhancement-Based Methods

Data enhancement can not only expand the number of training samples by adding image samples, but also realize the operation of data enhancement by enhancing the data of feature space.

Humans can easily recognize the category of objects in an image even when the image is deformed and some information is lost. Chen et al. [82] believe that although the deformed images may be visually unreal, they still retain key semantic information. Inspired by the latest progress in meta-learning, the Image Deformation Meta-Network (IDeMe-Net) is designed. The network combines the meta-learner with the image deformation sub-network to generate additional training examples and optimizes the two models in an end-to-end manner.

Dixit et al. [83] proposed a method for data augmentation by synthesizing sample features under the condition of expected attribute values in the attribute space: Attribute-Guided Augmentation (AGA). The model maps image samples into an attribute space and trains the encoder-decoder of the model to make the model generate images in different depths and poses. Schwartz et al. [84] proposed a data augmentation method Δ-encoder based on an improved auto-encoder. Specifically, an Auto-Encoder (AE) is used to extract the changes and differences between training category instance pairs, and then the differences are applied to a few samples of the new category to generate new samples. Finally, the classifier is trained with the expanded dataset. Chen et al. [85] proposed a semantic feature augmentation algorithm SFA, which uses the TriNet model based on an encoder-decoder to map samples to the semantic space, learn the concept of samples in the semantic space, expands samples in the semantic space by adding noise and finding nearest neighbors, and then map them back to the visual space, so as to obtain more expanded samples.

The current feature enhancement-based methods only deal with situations where there is only one category label in each image, while the multi-label situation has never been mentioned. Focusing on this problem, Alfassy et al. [86] proposed the Label-Set Operations network (LaSO) for multi-label few-shot image classification tasks, which uses

the relationship between label sets to extract potential semantic information to form a data augmentation at the feature space level.

### 3.4. Multimodal-Based Methods

Modality refers to the way people receive information, including hearing, sight, smell, touch, and many other ways [87]. Multimodal learning refers to the use of the complementarity between multiple modalities to eliminate the redundancy between the modalities, so as to learn a better feature representation. The existing few-shot image classification problems often focus on only single mode of image. However, in many problem scenarios, we only use a small amount of single-mode supervision information and ignore a large number of easily accessible information of other modes. The multimodal information of the target can provide more prior knowledge to make up for the lack of supervision information in the image data. Few-shot image classification, based on multimodal, mainly uses image information combined with text, speech, and other modal information to obtain more prior knowledge, so as to better complete the image classification tasks. With the development of deep learning, more and more scholars have begun to pay attention to the research of few-shot image classification based on multimodal. According to the different learning ways of multimodal, this paper divides it into two multimodal few-shot learning methods based on knowledge transfer and metric. Next, the typical algorithms and research progress of these two methods are introduced.

### 3.4.1. Knowledge Transfer-Based Methods

The few-shot learning task usually only contains dozens of categories, while the large-scale few-shot learning task contains thousands of classes of images with few sample images for each class, which also brings a lot of difficulties to the algorithm. In order to solve this problem, Li et al. [88] proposed a class hierarchical structure for predicting the affiliation of a certain sample class, using the semantic relationship between the source set and the target set class as a kind of prior knowledge to help the network learn more transferable feature information. This tree-shaped class hierarchical structure explicitly expresses the semantic relationship. The network structure of the model is shown in Figure 9.

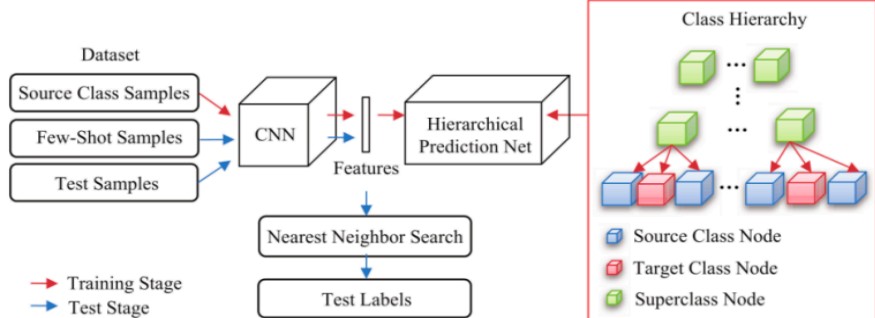

**Figure 9.** Overview of the knowledge transfer with class hierarchy network.

The model adopts CNN to extract the visual features of images, and then inputs them into the class hierarchy network to predict the classes in two stages: (1) directly input the classes of each layer in the fully connected layer. (2) Information from different layers are fused to predict classes. After training, the nearest neighbor algorithm is used to classify the feature vectors in the test stage.

The visual and semantic feature spaces have different structures according to their definitions. For some concepts, visual features may be richer and more discriminative than textual features. However, when visual information is limited in image classification, semantic representation can provide powerful prior knowledge and context to facilitate learning. In order to fully mine prior knowledge, Peng et al. [89] proposed the Knowledge

Transfer Network (KTN), which combines visual feature learning, knowledge inferring, and classifier learning into a unified few-shot image recognition framework. Among them, the knowledge transfer module adopts a graph convolution neural network. Each node represents the word vector corresponding to a class label, and the edge between nodes represents the correlation between the two classes. Finally, image features and semantic features are combined by classifier weight fusion, and the semantic features are added to the few-shot classifier as a priori knowledge.

### 3.4.2. Metric-Based Methods

Human children use a combination of semantic information to learn new things. Based on this idea, Schwartz et al. [90] proposed a few-shot learning algorithm that combines multiple semantic information. The algorithm is based on the idea of the prototypical network. First, CNN is used to extract the visual space prototype $V$; then, for a variety of semantic information such as semantic tags, image descriptions, and object attributes, the corresponding embedding network is used for feature extraction; next, the feature information is transformed into the corresponding semantic prototype $S_i$; finally, the semantic prototype and visual prototype are fused according to a certain weight to obtain the fusion prototype $P$. The similarity between the fusion prototype $P$ and the visual feature $Q$ of the query set image is measured, and the class label is predicted.

Under the condition of having few samples, sometimes the image feature information has higher discrimination, and sometimes the semantic information has more significant discrimination. In order to improve the classification accuracy, Xing et al. [91] proposed an Adaptive Modality Mixture Mechanism (AM3) based on the prototypical network. It introduces semantic feature information in the feature extraction stage to cooperate with the original visual prototype and uses an adaptive hybrid network to adjust the fusion ratio of semantic features and image features. Hence it can adaptively and selectively combine the information of the two modes for few-shot learning. The transformation mapping and adaptive coefficients are learned by the neural network. This method uses the mixed feature information to greatly improve the classification effect of the original algorithm.

### 3.5. Comparison of Different Learning Paradigms

In order to further analyze the advantages and disadvantages of different learning paradigms of few-shot image classification algorithms, this section compares these four classification methods in detail, as shown in Table 2.

**Table 2.** Comparison of advantages and disadvantages of few-shot image classification algorithms with different paradigms.

| Method | Characteristic | Advantage | Disadvantage |
| --- | --- | --- | --- |
| Transfer Learning | Transfer of the useful prior knowledge | Alleviate of overfitting | Negative transfer |
| Meta-Learning | Usage of prior knowledge to guide the learning of new tasks | Excellent performance | Complex model |
| Data Augmentation | Usage of auxiliary information to expand sample data | Prevention of overfitting | Poor generalization ability |
| Multimodal | Usage of the information of auxiliary modalities to classify images | Better feature representation | Hard to train and calculate |

In few-shot image classification tasks, transfer learning can achieve better accuracy by pre-training on large datasets in the source domain and fine-tuning on small datasets in the target domain. At the same time, transfer learning is also facing great challenges. When the dataset categories differ greatly, the classification accuracy of the model will be greatly reduced. In recent years, methods based on meta-learning have achieved good classification results and become the mainstream method in the field of few-shot image classification. This method solves the problem of how to make the model learn how to learn. However, meta-learning methods generally have problems such as complex models

and large amounts of calculations. Researchers can design simple and effective models to perform better in few-shot image classification tasks. Although the method based on data augmentation can expand the sample data and reduce the overfitting problem to a certain extent, due to the small amount of sample data, the transformation mode is limited. Although the training effect has been improved to a certain extent, it cannot completely solve overfitting. In recent years, with the development of multimedia technology, the multimodal technique has gradually become a current research hotspot. Methods based on multimodal can use the information of auxiliary modalities to improve the expression ability of image features, but it is easy to be disturbed by noise in the process of information fusion, which makes the fused information inaccurate. Therefore, how to study a more appropriate fusion method is the future development trend of multimodal methods.

## 4. Comparison of Different FSIC Algorithms

### 4.1. Quantitative Comparison of FSIC Algorithms

In order to compare the performance of different few-shot image classification algorithms, this section summarizes the experimental results of some representative algorithms on public datasets, together with analyzation and conclusions. At the same time, in order to explore the classification performance of few-shot image classification models on professional field data, several representative algorithms are selected for experimental comparison on the CIIP-TPID dataset applied in the field of public security. The classification accuracy (Accuracy) of 5-way 1-shot and 5-way 5-shot was used as the evaluation criteria. Accuracy is the evaluation criteria widely used by scholars in few-shot image classification tasks to evaluate the classification performance of the model. Classification accuracy refers to the proportion of the number of samples correctly classified by the model in the total number of samples.

$$\text{Accuracy} = \frac{\text{Number of correctly classified samples}}{\text{Number of total samples}} \times 100\%$$

4.1.1. Comparison of Performance on Benchmark Datasets

This section analyzes and compares the experimental results of some representative few-shot image classification algorithms on the two benchmark datasets, Omniglot and miniImageNet. These two datasets are widely used in the field of few-shot learning. The Omniglot dataset consists of character images and its content is relatively simple. Although the sample content of the miniImageNet dataset is complex, it is suitable for prototypical design and experimental research. The experimental results are shown in Table 3.

As can be seen from Table 3:

1.  The few-shot image classification algorithms have a high classification accuracy on the Omniglot dataset, while their performance on the miniImageNet dataset is relatively poor.
2.  The accuracy of the 5-way 5-shot task in the two datasets is higher than that of the 5-way 1-shot task, which shows that the more training data in the sample category, the more features the model can learn, which is conducive to improving the classification accuracy.
3.  On the Omniglot dataset, the accuracy of the selected algorithms is more than 98%, and the experimental results are slightly different; on the miniImageNet dataset, the experimental results of different algorithms differ greatly. The accuracy of the 5-way 1-shot task is mostly about 50%. The classification result of the GCN-NFO method with the best performance is about 55% higher than that of the Meta-Learner LSTM method. The classification accuracy of the 5-way 5-shot task is mostly about 65%. Among them, the GCN-NFO method, which has the best classification effect, improves by about 43% compared to the worst-performing MN method, indicating that the existing few-shot image classification algorithm can still make a great improvement on the miniImageNet dataset.

**Table 3.** Comparison of experimental results of few-shot image classification algorithms on Omniglot and miniImageNet.

| | 5-Way Accuracy (%) | | | |
|---|---|---|---|---|
| **Algorithm** | **Omniglot** | | **miniImageNet** | |
| | **1-Shot** | **5-Shot** | **1-Shot** | **5-Shot** |
| MTL [46] | - | - | $61.20 \pm 1.80$ | $75.50 \pm 0.80$ |
| SNAIL [47] | $99.07 \pm 0.16$ | $99.78 \pm 0.09$ | $55.71 \pm 0.99$ | $68.88 \pm 0.92$ |
| Meta-Learner LSTM [51] | - | - | $43.44 \pm 0.77$ | $60.60 \pm 0.71$ |
| MAML [54] | $98.7 \pm 0.4$ | $99.9 \pm 0.1$ | $48.70 \pm 1.84$ | $63.11 \pm 0.92$ |
| Meta-SGD [55] | $99.53 \pm 0.26$ | $99.93 \pm 0.09$ | $50.47 \pm 1.87$ | $64.03 \pm 0.94$ |
| TAML [56] | $99.5 \pm 0.3$ | $99.81 \pm 0.1$ | $51.73 \pm 1.88$ | $66.05 \pm 0.85$ |
| RN [58] | $99.6 \pm 0.2$ | $99.8 \pm 0.1$ | $50.44 \pm 0.82$ | $65.32 \pm 0.70$ |
| MN [59] | 98.1 | 98.9 | $43.56 \pm 0.84$ | $55.31 \pm 0.73$ |
| PN [60] | 98.8 | 99.7 | $49.42 \pm 0.78$ | $68.20 \pm 0.66$ |
| GNN [67] | 99.2 | 99.7 | $50.33 \pm 0.36$ | $66.41 \pm 0.63$ |
| EGNN [68] | 99.7 | 99.7 | 62.3 | 76.37 |
| TPN [69] | 99.2 | 99.4 | 55.51 | 69.86 |
| GCN-NFO [73] | 99.87 | 99.96 | 98.57 | 98.58 |
| MetaGAN [77] | $99.67 \pm 0.18$ | $99.86 \pm 0.11$ | $52.71 \pm 0.64$ | $68.63 \pm 0.67$ |
| PMN [80] | - | - | 57.6 | 71.9 |
| IDeMe-Net [82] | - | - | $59.14 \pm 0.86$ | $74.63 \pm 0.74$ |
| Δ-encoder [84] | - | - | 59.9 | 69.7 |
| AM3 [91] | - | - | $65.30 \pm 0.49$ | $78.10 \pm 0.36$ |

4.1.2. Comparison of Performance on Application Dataset

This section selects some representative algorithms and conducts experiments on the CIIP-TPID dataset. The experiments are tested on the three sub-datasets of surface pattern image (Surface), indentation pattern image (Indentation), and mixed pattern image (Mix), respectively. Among them, each category of the surface pattern image dataset and the indentation pattern image dataset contains 80 tire surface patterns and 80 tire indentation pattern images, respectively. The mixed pattern dataset contains 160 mixed images of surface pattern and indentation pattern per class. The experimental results are shown in Table 4.

It can be seen from Table 4:

1. The experimental results of various few-shot image classification algorithms on the CIIP-TPID dataset are relatively good, better than those on the miniImageNet dataset, but lower than those on the Omniglot dataset. The reason is that the miniImageNet dataset has a wide variety of image samples and complex content; the Omniglot dataset is composed of different handwritten character images with a single background and simple content; while the CIIP-TPID dataset consists of different types of tire pattern images and includes indentation images on different carriers. Its image background is slightly richer and the content is relatively simple.

2. The results of different methods on the mixed dataset are relatively low. This is because the mixed data contains two kinds of data: surface pattern image and indentation pattern image. The samples in each category are relatively complex with characteristics of large intra-class differences and small inter-class differences, which bring difficulties to the classification task.

3. GNN and GCN-NFO have the highest accuracy in the 5-way 1-shot task, indicating that the metric learning method based on the graph neural network is more suitable for the study of tire pattern image classification.

4. The GCN-NFO method achieves the best classification effect on different sub-datasets. This is because GCN-NFO makes full use of the image features of the special data samples to improve the network performance. In the next step, we will compare more algorithms and try to conduct more in-depth research on datasets in other fields.

**Table 4.** The experimental results of few-shot image classification representative algorithms on the CIIP-TPID dataset.

| Algorithm | Datasets | 5-Way Accuracy (%) | |
| --- | --- | --- | --- |
| | | **1-Shot** | **5-Shot** |
| Meta Networks [9] | Surface | 53.46 | 78.42 |
| | Indentation | 66.13 | 80.45 |
| | Mix | 42.80 | 63.53 |
| MAML [54] | Surface | 67.09 | 85.55 |
| | Indentation | 77.66 | 87.32 |
| | Mix | 46.03 | 64.00 |
| RN [58] | Surface | 63.97 | 81.60 |
| | Indentation | 73.71 | 84.54 |
| | Mix | 48.21 | 65.20 |
| GNN [67] | Surface | 77.46 | 89.52 |
| | Indentation | 77.76 | 92.00 |
| | Mix | 58.04 | 79.98 |
| GCN-NFO [73] | Surface | 89.12 | 94.04 |
| | Indentation | 95.84 | 88.14 |
| | Mix | 99.62 | 88.20 |
| SFA [85] | Surface | 72.71 | 91.03 |
| | Indentation | 76.42 | 91.76 |
| | Mix | 51.84 | 81.02 |

*4.2. Qualitative Comparison of FSIC Algorithms*

Based on the theoretical principle of the algorithm, this section compares the different methods in each paradigm of few-shot image classification algorithms, and then analyzes their advantages and disadvantages. The results are shown in Table 5.

**Table 5.** Comparative analysis of different methods in each paradigm of few-shot image classification algorithms.

| Category | Method | Advantage | Disadvantage |
| --- | --- | --- | --- |
| Transfer Learning | Instance-based | Easy to implement | Data distribution is often different |
| | Feature-based | Good feature selection and transformation | Prone to overfitting |
| | Fine-tuning-based | Alleviate overfitting | The number of iterations should be less |
| Meta-Learning | Model-based | Strong generalization | Complex model and extensive calculations |
| | Optimization-based | Make models learn new tasks quickly | Extensive calculations and high memory consumption |
| | Metric-based | Easy to calculate | Weak interpretability and high memory consumption |
| Data Augmentation | Data generation-based | Increase sample numbers | Cannot completely solve overfitting |
| | Feature enhancement-based | Increase feature numbers | Easy to be disturbed by noise |
| Multimodal | Knowledge transfer-based | Learn better feature representation | Easy to be disturbed by noise during the fusion process |
| | Metric-based | Simple calculation and high accuracy | Weak interpretability and high memory consumption |

## 5. Applications of Few-Shot Image Classification

The research on few-shot image classification has high practical application value in fields, such as medicine, public security, and commerce. The sample sets that need to be

identified in these research fields have the characteristics of small scale, few annotations, and rare categories, and many deep learning models with excellent performance cannot be properly applied in these fields. Few-shot image classification can effectively solve this problem, making these high-performance models more widely used in various fields. This section introduces the specific applications of few-shot image classification from the three fields of medicine, public security, and commerce.

## 5.1. Medical Field

Medical image classification has a very important application value and can provide a key basis for patient condition analysis. Due to the limited source of medical images and the manual processing of labeled data by experts, there is a lack of labeled training data. In addition, some samples are not publicly available because of their secrecy. The above factors lead to a small number of available samples for research, and the few-shot image classification technology is, therefore, suitable for medical image processing. At present, few-shot image classification has some practical applications in the medical field [92–95]. Figure 10 shows some image examples in the medical field.

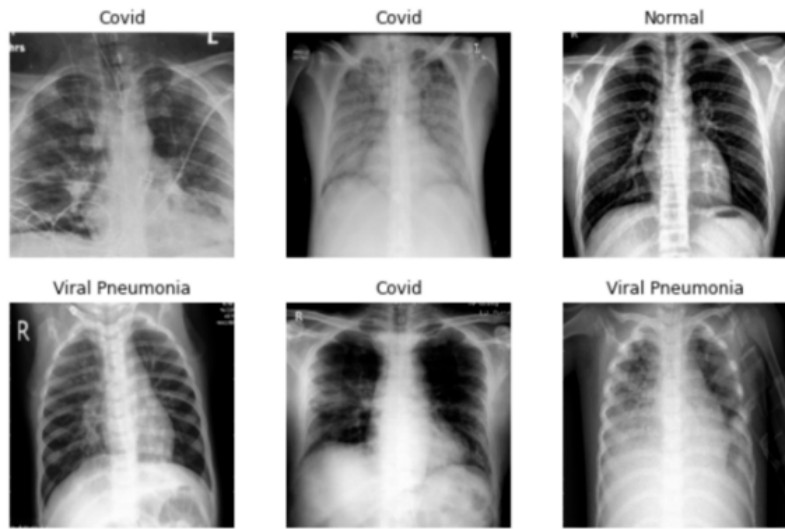

**Figure 10.** Image examples in the medical field.

Shang et al. [92] combined transfer learning, multi-task learning, and semi-supervised learning methods into a unified framework to promote a better performance of medical image classification. Cai et al. [93] proposed an end-to-end learning model combined with an attention mechanism to solve the problem of medical image classification and extract features from space and channels, so as to enhance the representation ability of the model.

Chen et al. [94] proposed a few-shot learning method for the automatic screening of COVID-19 images. An encoder is trained by comparative learning, which can capture the feature representation on the lung dataset and classify it by prototype network. Jadon [95] proposed a few-shot learning model for COVID-19 detection with a combination of a Siamese network and transfer learning, which ultimately provided excellent classification results.

At this stage, there are still some difficulties in fully applying few-shot image classification technology to the medical field. The reasons are threefold:

1.  There are subtle differences in medical images, which usually lead to certain recognition errors and make the model learn unnecessary features, and finally affect the classification results.
2.  Most medical images are 2D images, which cannot truly reflect the 3D structure information of the human body. This will lead to the loss of certain effective information in the process of collecting images, and finally, result in inaccurate classification results.

3. Analysis of medical images alone is not enough to accurately judge the disease but requires collaboration with multimodal technologies. Although the current few-shot image classification has made good achievements in the medical field, there is still a lot of research space in the future.

*5.2. Public Security Field*

Few-shot image classification has extremely important application value in the field of public security. Tire pattern image classification is an important way to provide key clues in the processing of traffic cases and the detection of criminal cases. In addition, as an important biological feature, palmprint also contains a wealth of information. Not only fingerprints and shoe prints will be left at the crime scene, but sometimes incomplete palmprint information will be left as well, which can provide important clues for case detection. Therefore, palmprint recognition also has a very important application value in the field of public security. Figure 11 shows some image examples in the public security field.

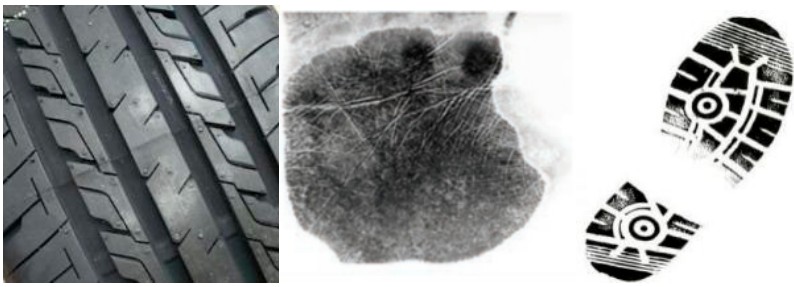

**Figure 11.** Image examples in the public security field.

Tire pattern images have complex texture, single color, and similar visual effects of different types of tire patterns, which bring great difficulties to the classification task. In order to improve the classification accuracy of tire pattern images, Liu et al. [96] proposed a feature extractor using the CNN model based on the idea of transfer learning. The pre-processed model on ImageNet is applied to the tire pattern image dataset, and then the model parameters are fine-tuned to enhance the representation ability of the new model. In [97], a feature fusion algorithm, based on transfer learning, is proposed. The algorithm transfers the pre-trained CNN model to the tire pattern image dataset through transfer learning and fine-tuning of model parameters. In addition, in order to further improve the classification performance, the obtained CNN features and low-level image features are combined as fusion features to train the SVM classifier, which improves the classification effect of the tread pattern image.

Shao et al. [98] proposed a few-shot palmprint recognition method based on the graph neural network. The palmprint features extracted by the convolutional neural network are processed as nodes in the GNN. The edges of the graph network represent the similarity between the graph nodes. By continuously optimizing the parameters in the network, the category of image samples is finally predicted.

In the field of public security, due to the problems of security and secrecy, it is often difficult to obtain a large number of publicly used image data. This results in the lack of large-scale annotation samples for research, which easily leads to overfitting in classification tasks. At the same time, many criminal investigation images often have problems such as complex backgrounds and unclear images. In the process of image classification, it is often necessary to combine image denoising, image enhancement, image clarity, and other techniques. In addition, most of the current criminal investigation images are two-dimensional planar images obtained by cameras, which cannot truly reflect the three-dimensional information of the scene. This will lose certain key information and make the final classification results be inaccurate. Therefore, there is still much room for improvement in applying few-shot image classification to the public security field.

### 5.3. Commercial Field

With the increasing maturity of artificial intelligence, commodity information recognition has become a promising application direction. In the commodity recognition scenario, there are often problems of a serious shortage of commodity sample data and unbalanced data distribution. The use of general image classification methods will result in lower accuracy for categories with fewer samples. Therefore, how to use the machine learning model to learn effectively in the case of few-shots is an important research direction to realize the commercialization of commodity classification. At present, few-shot image classification has been successfully applied in the commercial field, such as commodity image classification, bank operational risk classification, and so on. Figure 12 shows some examples of the commodity image.

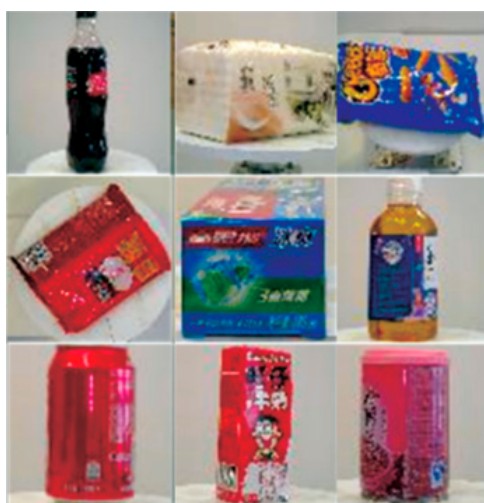

**Figure 12.** Examples of the commodity image.

In the field of commodity image recognition, there are often problems such as numerous image categories, similar features, and scarcity of datasets. The recognition capabilities of traditional deep learning models are limited. Xu et al. [99] proposed a Local Descriptor Relation Network (LDR-Net) by combining the idea of deep local descriptors and metric learning for supermarket retail commodity image classification. Lu et al. [100] proposed an improved residual network model to improve the classification performance of clothing images. By improving the arrangement order of "BN + ReLU + Convolutional Layer" in the traditional residual block, the work introduces the attention mechanism and adjusts the network convolution kernel structure, and the recognition and classification effect of the model is improved.

Wu et al. [101] proposed a few-shot learning method for the commercial field–Probabilistic Network, which improves the sensitivity of the model to commerce data by means of data augmentation. The generalization ability of this method is verified by studying two cases of absenteeism prediction and online hotel reviews.

At present, the core challenge of commodity information recognition is how to subdivide the various and extremely similar commodity categories in the market. In addition, there is no guarantee that it will have the same efficient and accurate recognition ability for new products that are constantly appearing. At the same time, most of the existing few-shot classification models only use the modal information of images to classify commodities and do not make full use of other modal information, resulting in inaccurate classification results. Therefore, it is a research trend in the field of commodity information recognition in the future to integrate text, voice, and other modal information to complete the classification task. In addition, commodity data also has problems, such as unbalanced data distribution. To fully utilize the few-shot image classification in the commercial field it is also necessary to consider collecting datasets with wider categories and richer contents.

## 6. Research Trend of Few-Shot Image Classification

Few-shot image classification has always been a critical and difficult problem in image classification since it was proposed. Although the models and algorithms proposed in the field of few-shot learning have improved the accuracy of image classification to varying degrees, at present, there are still some obstacles to few-shot image classification. Difficulties, such as limited dataset quality, artificially designed neural network architectures, weak interpretability of neural network, multimodal fusion, etc., need to be further studied urgently. This section focuses on the current research difficulties of few-shot image classification and discusses future research trends.

### 6.1. Build Suitable Datasets for Practical Applications

Although there are currently some datasets for few-shot image classification tasks, there are still some deficiencies in both the number of image categories and the quality of labeling. They generally have the characteristics of prominent foreground objects and single background [102], which are not common images in practical application. The few-shot image classification is a research topic combined with practical applications. Its performance is closely related to the scale and quality of the dataset. The richer the training images, the more obvious the performance improvement and the stronger the practicality. To make few-shot image classification widely used in real life, we have to consider the image recognition problems of complex scenes such as illumination, blur, occlusion, and low resolution. In addition, the existing few-shot image classification methods are mostly aimed at universal datasets. Due to the data security requirements and the difficulty of data collection, there are relatively few pieces of research on data in special fields. Therefore, how to construct larger-scale, higher-quality image datasets and produce dedicated datasets for different fields is an important research problem in few-shot image classification.

### 6.2. Neural Architecture Search

In recent years, deep learning models have achieved good results on few-shot image classification tasks, but the numerous hyperparameters and network structure parameters that follow will produce explosive combinations. The efficiency of conventional random search and grid search is very low. In addition, the current design of neural network architecture still depends on manual work to a large extent, which is time consuming, laborious and error prone. Therefore, how to transform the neural network architecture from the manual design to automatic machine design has become a problem that researchers pay attention to. Neural Architecture Search (NAS) [103,104] can solve this problem. At this stage, scholars have applied the NAS method to image classification tasks [105,106], and it is better than manually designed architectures. Therefore, how to design a better network architecture than manual design through NAS, and how to expand NAS method to other deep learning related hyperparametric optimization, so as to obtain better learning parameters than manual design, is still a major research hotspot in the field of few-shot image classification in the future.

### 6.3. Interpretability of Neural Networks

Although neural networks have played a key role in various fields, their limited interpretability has always been a problem, which is what we often call the "black box problem". Even if we try to generalize the limited training data to unknown inputs, they may fail in the case of small interference. Moreover, this practice will make it difficult to verify the robustness of the algorithm. In recent years, there has been more and more work hoping to explore what the neural network has learned [107–110]. Huang et al. [108] proposed an interpretable depth model for fine-grained image classification, which increased the interpretability of the model. Selvaraju et al. [109] proposed a novel class discrimination positioning technology Grad-CAM (Gradient-weighted Class Activation Mapping), which generates visual interpretation to make any CNN-based models more transparent. How-

ever, how to propose a neural network with strong interpretability is still a research hotspot in the field of few-shot image classification and even the whole field of deep learning.

### 6.4. Multimodal Few-Shot Image Classification

With the rapid development of information technology, the multimodal technique has gradually become a research hotspot. Multimodal deep learning has brought great opportunities and challenges to machine learning. At present, some scholars have applied the multimodal technique to few-shot image classification [88–91]. For example, the KTN network proposed in [89] combines the image features and semantic features for few-shot image classification tasks; the AM3 method proposed in [91] can adaptively and selectively combine semantic features and visual features, which greatly improves the classification effect of the original algorithm. These methods can not only effectively avoid the overfitting problem in the learning process, but also improve the classification effect to a certain extent. However, the process of multimodal information fusion is easy to be disturbed by noise, which makes the fused information inaccurate. Therefore, how to develop a more appropriate multimodal fusion method to further improve the classification effect is a research trend of few-shot image classification.

## 7. Conclusions

According to the different mechanisms, this paper divides the existing few-shot image classification methods into four learning paradigms: transfer learning-based, meta-learning-based, data augmentation-based, and multimodal-based. Transfer learning can transfer the useful prior knowledge from the source domain to the target domain, which is conducive to few-shot learning; meta-learning employs the prior knowledge learned from a large number of tasks to make the model learn how to learn; methods based on data augmentation improve the diversity of samples by generating data or enhancing the number of features in the feature space; the multimodal method uses the complementarity between multiple elements of modal information and eliminates the redundancy between the modalities, so as to learn better feature representation. In addition, this paper summarizes the commonly used datasets for few-shot image classification and analyzes the advantages and disadvantages of different algorithms. Then, the application of few-shot image classification in several fields is analyzed. In the end, based on the theoretical research results and practical applications of few-shot image classification, this paper summarizes several future research trends in this field.

**Author Contributions:** Conceptualization, Y.L.; methodology, Y.L.; software, H.Z.; validation, H.Z.; formal analysis, H.Z.; investigation, H.Z.; resources, Y.L.; data curation, H.Z.; writing—original draft preparation, H.Z.; writing—review and editing, Y.L., W.Z., G.L., Q.T. and N.L.; supervision, Y.L., W.Z., G.L., Q.T. and N.L.; project administration, Y.L. and W.Z.; funding acquisition, H.Z. and W.Z. All authors have read and agreed to the published version of the manuscript.

**Funding:** This research was funded by National Natural Science Foundation of China, grant number 62106195, in part by the Graduate Innovation Fund Project of Xi'an University of Posts and Telecommunications, grant number CXJJDL2021013.

**Data Availability Statement:** Data sharing is not applicable to this article as no datasets were generated or analyzed during the current study. Data for this research are unavailable.

**Conflicts of Interest:** The authors declare no conflict of interest.

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
