# Peer review of "Few-Shot Image Classification: Current Status and Research Trends"

_electronics, doi:10.3390/electronics11111752_

Round 1

Reviewer 1 Report

The paper reviews few-shot image classification and points out the future directions on this field. It would be interesting to the machine learning community. However, some more references could be added to explain the motivation of this area and categorization of this family of algorithms. For example, at line 39 p.1, it needs to specify under which situations the classical methods do not produce reliable results, together with some references. At line 97, the references for the term "N-way K-shot problem" should be included. In Table 1, the image datasets are grouped into simple, complex and special ones, which lacks the explanation. In other words, how would a given dataset be considered as simple/complex/special? Furthermore, since it's a review paper, it would be great to see the difference and similarity of various algorithms, especially their respective motivations. More comparison tables could be added. This will be particularly useful for the audience to pick up an appropriate method for a specific problem. Finally, comparison of computational complexity or cost should also be included. 

Reviewer 2 Report

As the authors claimed the contributions of their work are: 1) dividing the existing few-shot image classification algorithms into four categories: transfer learning-based, meta-learning-based, data augmentation-based, and multimodal-based. 2) list of dataset. 3) Performance comparison 4) A list of future research directions in few-shot image classification is identified

1- My main comment is on the way they categorized the papers into 4 distinct categories and in each category again they sub-categorized into some areas. I don't think it is too rigid as they categorized it. It is somewhat overlapping thus the best way to organize the papers would be via Venn diagram as opposed to a simple categorical chart. There are some methods that fall very clearly in the intersection of augmentation and TL for example. Or there are some of them that fall into the common intersection of multi-modal TL.

2- Also, categorizing the sections into 3.2.1. Model-Based Methods and 3.2.2. Optimization-Based Methods do not make much sense to me as any model needs optimization and the titles are misleading. I understand that categorization is done for better understanding but being too rigid, makes things inaccurate, I believe. So, I would suggest using overlapping thus a more realistic categorization.
3- Also, image classification on its own is too broad. I think if they can orient it to health applications, for example, their paper would be more useful to the readers.
